# *N*,*N*-Bis(2,4-Dibenzhydryl-6-cycloalkylphenyl)butane-2,3-diimine–Nickel Complexes as Tunable and Effective Catalysts for High-Molecular-Weight PE Elastomers [note 1]

**DOI:** 10.3390/molecules28124852

**Published:** 2023-06-19

**Authors:** Shu Jiang, Yuting Zheng, Irina V. Oleynik, Zhixin Yu, Gregory A. Solan, Ivan I. Oleynik, Ming Liu, Yanping Ma, Tongling Liang, Wen-Hua Sun

**Affiliations:** 1School of Pharmaceutical Sciences, Changchun University of Chinese Medicine, Changchun 130117, China; 2Key Laboratory of Engineering Plastics, Beijing National Laboratory for Molecular Sciences, Institute of Chemistry, Chinese Academy of Sciences, Beijing 100190, China; 3Vorozhtsov Novosibirsk Institute of Organic Chemistry, Pr. Lavrentjeva 9, Novosibirsk 630090, Russia; 4Department of Chemistry, University of Leicester, University Road, Leicester LE1 7RH, UK

**Keywords:** nickel, polyethylene elastomers, high activity, high molecular weight, good control, activator effects, mechanical properties

## Abstract

Four examples of *N*,*N*-bis(aryl)butane-2,3-diimine–nickel(II) bromide complexes, [ArN=C(Me)-C(Me)=NAr]NiBr_2_ (where Ar = 2-(C_5_H_9_)-4,6-(CHPh_2_)_2_C_6_H_2_ (**Ni1**), Ar = 2-(C_6_H_11_)-4,6-(CHPh_2_)_2_C_6_H_2_ (**Ni2**), 2-(C_8_H_15_)-4,6-(CHPh_2_)_2_C_6_H_2_ (**Ni3**) and 2-(C_12_H_23_)-4,6-(CHPh_2_)_2_C_6_H_2_ (**Ni4**)), disparate in the ring size of the *ortho*-cycloalkyl substituents, were prepared using a straightforward one-pot synthetic method. The molecular structures of **Ni2** and **Ni4** highlight the variation in the steric hindrance of the *ortho*-cyclohexyl and -cyclododecyl rings exerted on the nickel center, respectively. By employing EtAlCl_2_, Et_2_AlCl or MAO as activators, **Ni1**–**Ni4** displayed moderate to high activity as catalysts for ethylene polymerization, with levels falling in the order **Ni2** (cyclohexyl) > **Ni1** (cyclopentyl) > **Ni4** (cyclododecyl) > **Ni3** (cyclooctyl). Notably, cyclohexyl-containing **Ni2**/MAO reached a peak level of 13.2 × 10^6^ g(PE) of (mol of Ni)^−1^ h^−1^ at 40 °C, yielding high-molecular-weight (ca. 1 million g mol^−1^) and highly branched polyethylene elastomers with generally narrow dispersity. The analysis of polyethylenes with ^13^C NMR spectroscopy revealed branching density between 73 and 104 per 1000 carbon atoms, with the run temperature and the nature of the aluminum activator being influential; selectivity for short-chain methyl branches (81.8% (EtAlCl_2_); 81.1% (Et_2_AlCl); 82.9% (MAO)) was a notable feature. The mechanical properties of these polyethylene samples measured at either 30 °C or 60 °C were also evaluated and confirmed that crystallinity (*X*_c_) and molecular weight (*M*_w_) were the main factors affecting tensile strength and strain at break (*ε*_b_ = 353–861%). In addition, the stress–strain recovery tests indicated that these polyethylenes possessed good elastic recovery (47.4–71.2%), properties that align with thermoplastic elastomers (TPEs).

## 1. Introduction

Polyethylene is a general-purpose resin that can adopt many types of structure, anywhere from linear (high-density) to highly branched (low-density) structures, with the result that it can display a myriad of different mechanical properties [1]. In the field of ethylene polymerization, late-transition metal-mediated processes have garnered considerable attention, as such catalysts can be readily tuned by performing steric/electronic changes to the supporting chelating ligand, the type of metal center and their coordination environment [2,3,4]. Of particular note is the α-diimine family of nickel and palladium catalysts pioneered by Brookhart and co-workers, which have the ability to mediate the formation of branched polyethylene by using ethylene as the sole feed in polymerization. Central to the function of these Group 10 catalysts is their unique ability to promote a process known as chain walking leading to branch formation [5,6,7,8,9,10,11,12,13,14].

With particular reference to nickel catalysts, numerous previous studies have focused on modifying the α-diimine ligand backbone or changing the stereo/electronic properties of the *N*-aryl substituents. Indeed, these studies have culminated in significant improvements in the activity and thermal stability of α-diimine–nickel catalysts [15,16]. Elsewhere, our team and others have been interested in using such catalysts to generate polyethylenes with high branching density along with mechanical properties characteristic of thermoplastic elastomers (TPEs). Amongst the α-diimine ligand frameworks to be employed in this area, *N*,*N*-bis(aryl)butane-2,3-diimine and bis(imino)acenaphthene (**A** in [Fig molecules-28-04852-ch001]) have been studied the most intensively [5,17,18,19,20,21].

As concerns the *N*,*N*-bis(aryl)butane-2,3-diimine class, various developments have been disclosed. For example, **B**-type precatalysts ([Fig molecules-28-04852-ch001]) bearing relatively small *ortho*-alkyl substituents (R^1^ and R^2^) on both *N*-aryl groups of the chelating ligand display, following suitable activation, good performance characteristics in ethylene polymerization (activity ≤ 3.01 × 10^6^ g(PE) of (mol of Ni)^−1^ h^−1^) [22], with an inclination towards generating high-molecular-weight polyethylene (*M*_w_ range: 2.22–8.97 × 10^5^ g mol^−1^) exhibiting low branching density (18 branches per 1000 Cs; mainly Me branches). By contrast, **C**-type nickel precatalysts ([Fig molecules-28-04852-ch001]) incorporating bulkier benzhydryl (CHPh_2_) groups at both *ortho*-positions display slightly higher-molecular-weight polyethylene (*M*_w_ range: 4.4–11.8 × 10^5^ g mol^−1^ [18]; 8.0–13.1 × 10^5^ g mol^−1^ [23]), but lower catalytic activity (TOF = 2029–2578 h^−1^ [18], 0.98–1.58 × 10^6^ g(PE) of (mol of Ni)^−1^ h^−1^ [23]). More noticeably, the branching density of polymers produced using **C** is higher (e.g., 63–75 branches per 1000 Cs [18]; 58 branches per 1000 Cs [23]) than that seen with **B**.

As an extension to our research program looking at **A**–**C**, we now disclose a series of *N*,*N*-bis(aryl)butane-2,3-diimine–nickel precatalysts (**Ni1**–**Ni4** in [Fig molecules-28-04852-ch001]) that contain cycloalkyl substituents in combination with benzhydryl groups at the *ortho*-positions of the *N*-aryl groups. These carbocyclic groups were targeted to explore how the systematic modification of the ring size and flexibility (i.e., cyclopentyl, cyclohexyl, cyclooctyl, and cyclododecyl) can impact the catalytic performance and structural characteristics (e.g., branching density and composition) of polyethylene. Furthermore, stress–strain and dynamic mechanical analyses were performed on selected samples to evaluate their mechanical and elastic properties and to investigate how the polymerization conditions, such as activator type and operating temperature, can play a role. Full synthetic and characterization data of all new nickel complexes are additionally reported.

## 2. Results

### 2.1. Synthesis and Characterization

The *N*,*N*-bis(aryl)butane-2,3-diimine–nickel(II) bromide complexes, [ArN=C(Me)-C(Me)=NAr]NiBr_2_ (where Ar = 2-(C_5_H_9_)-4,6-(CHPh_2_)_2_C_6_H_2_ (**Ni1**), Ar = 2-(C_6_H_11_)-4,6-(CHPh_2_)_2_C_6_H_2_ (**Ni2**), 2-(C_8_H_15_)-4,6-(CHPh_2_)_2_C_6_H_2_ (**Ni3**) and 2-(C_12_H_23_)-4,6-(CHPh_2_)_2_C_6_H_2_ (**Ni4**)), were prepared using an *in situ* one-pot method involving the reaction of 2,3-butanedione, the corresponding 2-cycloalkylaniline and (DME)NiBr_2_ (DME = 1,2-dimethoxyethane) in acetic acid at reflux (Figure 1) [24,25]. On work-up, **Ni1**–**Ni4** were isolated as brown powders in reasonable yields (17–64%) and characterized with FT-IR spectroscopy, elemental analysis and, in the case of **Ni2** and **Ni4**, single-crystal X-ray diffraction.

Single crystals of **Ni2** and **Ni4** suitable for X-ray determination were grown using slow diffusion as described in the experimental section. Perspective views of each are shown in Figure 1 and Figure 2; selected bond lengths and angles are listed in Table 1.

The structures of both **Ni2** and **Ni4** comprise a single nickel center that is surrounded by two nitrogen atoms from the bidentate *N*,*N*-bis(aryl)butane-2,3-diimine (aryl = 2-cyclohexyl-4,6-dibenzhydrylphenyl (**Ni2**) and 2-cyclododecyl-4,6-dibenzhydrylphenyl (**Ni4**)) and two bromide ligands to complete a geometry that can be best described as distorted tetrahedral; similar geometries have been previously reported for this class of *N*,*N*-nickel(II) complexes [18,19,26,27]. Some modest variation in the N1-Ni1-N2 bite angles in each is apparent, with values of 81.36(4)° (**Ni2**) and 80.60(2)° (**Ni4**), while the disparity in the Br1-Ni1-Br2 angles is slightly more distinct (123.970(11)° (**Ni2**) *vs.* 125.57(5)° (**Ni4**)). The Ni-N distances (2.0221(11), 2.0081(10) Å (**Ni2**) *vs.* 2.014(5), 2.004(5) Å (**Ni4**)) and the Ni-Br distances (2.3331(3), 2.3418(3) Å (**Ni2**) *vs.* 2.3304(13), 2.3361(13) Å (**Ni4**)) show minimal differences and are indeed comparable to those found in previous reports [18,19,26]. Similarly, the imine C=N bond lengths within the chelating ligand are typical for this functional group (range: 1.280(8)–1.2856(17) Å) [18,19,26,27]. The *N*-aryl rings are inclined towards the perpendicular with respect to the neighboring imine vectors (**Ni2** (88.48°, 87.00°) and **Ni4** (82.55°, 79.78°)), with the two *ortho*-cycloalkyl groups (and *ortho*-benzhydryl groups) being positioned on opposite sides of the diimine coordination plane. The carbocyclic rings themselves are puckered with the cyclohexyl ring in **Ni2**, adopting a chair conformation, while the larger cyclododecyl ring in **Ni4** can be best viewed as assuming a boat–chair–boat conformation. There are no intermolecular contacts of note.

### 2.2. Ethylene Polymerization Studies

#### 2.2.1. Selection of Aluminum Activator

With the aim to determine the most suitable activator that can afford the highest catalytic activity for the nickel precatalyst, cyclohexyl-substituted **Ni2** was chosen as the test precatalyst. Five different aluminum-alkyl reagents were initially screened, namely, Et_2_AlCl (diethylaluminum chloride), EtAlCl_2_ (ethylaluminum dichloride), EASC (ethyl sesquichloride), MAO (methylaluminoxane) and MMAO (modified methylaluminoxane). A 30 min polymerization run was performed with each **Ni2**/activator combination, with toluene being employed as reaction solvent and ethylene pressure being maintained at 10 atm; the Al:Ni ratios employed for each aluminum alkyl were based on typical values described elsewhere [22,23,28]. The results of this preliminary screening are listed in Table 2.

The inspection of the results reveals that all five aluminum alkyls were able to successfully activate **Ni2** but with varying levels of effectiveness: EtAlCl_2_ > Et_2_AlCl > MAO > MMAO > EASC. Based on the superior performance displayed by EtAlCl_2_, Et_2_AlCl and MAO, these three activators were selected for more in-depth evaluation (see below).

#### 2.2.2. Ethylene Polymerization Achieved with Activation of **Ni1**–**Ni4** with EtAlCl_2_

In order to optimize the performance of **Ni2**/EtAlCl_2_ and to establish an effective set of conditions that can be used to screen the other three nickel precatalysts, the effect of Al:Ni molar ratio, run temperature and reaction time were all systematically studied; the complete set of results is shown in Table 3.

With the Al:Ni molar ratio of **Ni2**/EtAlCl_2_ set at 500:1, the polymerization temperature was increased in ten-degree increments from 20 °C to 70 °C (runs 1–6; Table 3). Initially, the activity was seen to rise with the temperature, reaching the maximum activity of 10.3 × 10^6^ g(PE) of (mol of Ni)^−1^ h^−1^ at 30 °C (run 2; Table 3). However, at temperatures above 30 °C, some loss in performance was observed (runs 3–6; Table 3), which can be accredited to the partial deactivation of the active species and/or a decrease in the solubility of the ethylene monomer at higher temperatures [18,19,26,27,29]. Nonetheless, an appreciable level of performance was still maintained at 60 °C (6.60 × 10^6^ g(PE) of (mol of Ni)^−1^ h^−1^), highlighting the good thermostability of this catalyst (run 5; Table 3). In terms of the molecular weight of the polymer, this decreased gradually over the temperature range, while its dispersity became steadily narrower (*M_w_*/*M_n_*: 2.9 to 2.6). Indeed, the highest molecular weight was reached at 20 °C (12.5 × 10^5^ g mol^−1^), while at 60 °C and 70 °C, the molecular weight dropped to 4.64 × 10^5^ g mol^−1^ and 3.89 × 10^5^ g mol^−1^, respectively. These temperature effects were further borne out by their GPC traces (Appendix A) and can be attributed to thermally induced chain transfer [29,30,31,32]. As a further point, the melting points of the polymers (*T*_m_ range of 104.6–44.5 °C) tended to decrease with the increase in polymerization temperature, a finding that is expected to derive from molecular weight and branching variations (see below). 

Next, we set about determining the optimum Al:Ni molar ratio of **Ni2**/EtAlCl_2_ to promote the highest catalytic activity. Based on the above activity/temperature findings, the polymerization temperature was fixed at 30 °C, and the Al:Ni molar ratio was gradually increased from 300:1 to 700:1 (runs 2 and 7–10; Table 3). At 500:1, the activity attained an uppermost value of 10.3 × 10^6^ g(PE) of (mol of Ni)^−1^ h^−1^ (run 2; Table 3) and then, at higher ratios, showed a modest decrease, dropping to 9.12 × 10^6^ g(PE) of (mol of Ni)^−1^ h^−1^ at 700:1 (run 10; Table 3). More obviously, the activity increased when the Al:Ni molar ratio was adjusted from 300:1 to 500:1 (runs 2, and 7 and 8; Table 3), an observation that points towards a critical amount of EtAlCl_2_ being needed to induce good activity. As previously observed, the relative amount of EtAlCl_2_ used can affect the molecular weight of the polymer [33,34,35]. In this case, when the Al:Ni ratio varied between 300:1 and 700:1, the molecular weights of the resulting polymers were all high, with their levels ranging from 5.71 × 10^5^ g mol^−1^ at 300:1 to 7.08 × 10^5^ g mol^−1^ at 600:1 (Appendix A). The dispersity of the polymers was consistently narrow (*M_w_*/*M_n_* range: 2.4–3.1), in line with reasonable control of polymerization. 

The time/activity profile of **Ni2**/EtAlCl_2_ was then explored with the polymerization temperature being kept at 30 °C, and the amount of EtAlCl_2_, at 500 equivalents. By employing set run times of 5 min, 15 min, 30 min, 45 min and 60 min, it was observed that the activity reached a maximum after 5 min of 10.9 × 10^6^ g(PE) of (mol of Ni)^−1^ h^−1^ and then slowly decreased to 9.24 × 10^6^ g(PE) of (mol of Ni)^−1^ h^−1^ after 60 min (runs 2 and 11–14; Table 3). This observation is consistent with a rapid induction period for precatalyst activation, which was then followed by a relatively modest loss in activity over time (~15%), which highlights the sizable lifetime of the active catalyst. By contrast, the molecular weight of the polymer increased with time from 3.1 × 10^5^ g mol^−1^ after 5 min to 12.1 × 10^5^ g mol^−1^ after 60 min (Appendix A), while the dispersity remained relatively narrow (*M_w_*/*M_n_* range of 2.3–2.8), underlining the good control over time exhibited by this catalyst. 

To examine the effect of ring size variations in the *ortho*-cycloalkyl group on catalytic performance, the remaining three nickel(II) bromide complexes (**Ni1**, **Ni3** and **Ni4**) were evaluated for ethylene polymerization using the optimized reaction conditions identified using **Ni2**/EtAlCl_2_ (viz., Al:Ni molar ratio = 500:1, run temp. = 30 °C and run time = 30 min); the complete set of data including those on **Ni2** are shown in Table 3 (runs 2 and 17–19). By analyzing the data, it is evident that all four nickel complexes showed high to moderate catalytic activity (range: 10.3 × 10^6^ g(PE) of (mol of Ni)^−1^ h^−1^ to 0.26 × 10^6^ g(PE) of (mol of Ni)^−1^ h^−1^), with the relative order being **Ni2** (cyclohexyl) > **Ni1** (cyclopentyl) > **Ni4** (cyclododecyl) > **Ni3** (cyclooctyl). Based on these findings, it is evident that the steric factors exerted by the particular *ortho*-cycloalkyl group affected the catalytic activity, with the six-membered ring proving optimal. Furthermore, the flexibility of the *ortho*-cycloalkyl ring also played a role, as evidenced by 12-membered ring-containing **Ni4** displaying higher catalytic activity than 8-membered **Ni3** [9]. As for the polyethylenes generated, a noticeable variation in the molecular weight of polyethylenes produced with **Ni1**–**Ni4** could be seen, with values ranging from 4.37–8.22 × 10^5^ g mol^−1^ (Appendix A), with **Ni2** delivering the top-end value. Good control was again apparent for all four catalysts, with polymer dispersity values (*M_w_*/*M_n_*) between 2.1 and 2.9 being found.

#### 2.2.3. Ethylene Polymerization Achieved with Activation of **Ni1**–**Ni4** with Et_2_AlCl

To investigate the impact of the aluminum activator on both the performance of the nickel precatalyst and on the polyethylene properties, the study was extended to include Et_2_AlCl. Using an optimization approach similar to that performed using EtAlCl_2_, **Ni2** was again used as the precatalyst; the results of the evaluation using Et_2_AlCl are shown in Table 4.

As in the previous study employing **Ni2**/EtAlCl_2_, the temperature of the polymerization using **Ni2**/Et_2_AlCl was adjusted in this case from 30 °C to 70 °C, with the Al:Ni molar ratio being set at 500:1 (runs 1–5; Table 4). The results reveal that the maximum activity of 10.7 × 10^6^ g(PE) of (mol of Ni)^−1^ h^−1^ was seen at 40 °C rather than 30 °C for **Ni2**/EtAlCl_2_. Indeed, at 30 °C, **Ni2**/Et_2_AlCl and **Ni2**/EtAlCl_2_ displayed comparable performance (10.3 × 10^6^ g(PE) of (mol of Ni)^−1^ h^−1^ EtAlCl_2_ *vs.* 10.1 × 10^6^ g(PE) of (mol of Ni)^−1^ h^−1^ Et_2_AlCl). On the other hand, at 50 °C, the catalytic activity of **Ni2**/Et_2_AlCl started to show some noticeable loss in activity (7.16 × 10^6^ g(PE) of (mol of Ni)^−1^ h^−1^), which is consistent with the onset of partial deactivation of the active species and/or the lower solubility of ethylene at higher temperatures. Nonetheless, at 60 or 70 °C, further loss in activity was minimal, with a level of 6.40 × 10^6^ g(PE) of (mol of Ni)^−1^ h^−1^ being noted at 70 °C. By comparison, **Ni2**/EtAlCl_2_ exhibited a more significant decline in performance between 50 °C and 70 °C (from 9.24 × 10^6^ g(PE) of (mol of Ni)^−1^ h^−1^ to 2.26 × 10^6^ g(PE) of (mol of Ni)^−1^ h^−1^), which highlights apparent variations in catalyst thermal stability. 

In terms of the influence of run temperature on polymer molecular weight using **Ni2**/Et_2_AlCl, it was evident that the molecular weight of polyethylene fell as the temperature increased (from 9.98 to 5.09 × 10^5^ g mol^−1^). This downward trend was particularly noticeable at run temperatures between 50 °C and 60 °C, where *M_w_* dropped from 7.97 × 10^5^ g mol^−1^ to 5.50 × 10^5^ g mol^−1^ (Appendix A). Furthermore, some modest narrowing of the dispersity of the polymer was seen with the increase in temperature (*M_w_*/*M_n_*: from 2.5 to 2.2). 

To understand how the Al:Ni molar ratio affected the performance of **Ni2**/Et_2_AlCl, this was increased from 300:1 to 700:1, with the run temperature being set at 40 °C, and the run time, at 30 min (runs 2 and 6–9; Table 4). The highest polymerization activity was obtained at 500:1 (10.7 × 10^6^ g(PE) of (mol of Ni)^−1^ h^−1^). Moreover, it was apparent that the polymerization activity dramatically increased from 3.24 × 10^6^ g(PE) of (mol of Ni)^−1^ h^−1^ to 9.22 × 10^6^ g(PE) of (mol of Ni)^−1^ h^−1^ when the ratio was raised from 300:1 to 400:1, emphasizing the critical amount of activator required; similar findings are noted above and in the previous literature [36,37]. It is also noteworthy that the polymerization activity seen at 700:1 (5.54 × 10^6^ g(PE) of (mol of Ni)^−1^ h^−1^) was significantly lower than that at 600:1 (9.74 × 10^6^ g(PE) of (mol of Ni)^−1^ h^−1^), which indicates that excessive amounts of Et_2_AlCl can deactivate the catalyst. All the resulting polymers displayed high molecular weights ranging from 5.43 × 10^5^ g mol^−1^ to 8.85 × 10^5^ g mol^−1^ (Appendix A), with molar ratios in excess of 500:1 tending to form lower-molecular-weight polyethylene; similar observations have been reported [38,39]. In terms of polymer dispersity (*M_w_*/*M_n_*), the values were found between 2.4 and 3.3 but with no clear, identifiable trends. 

Polymerization runs using **Ni2**/Et_2_AlCl were then performed over set time periods of 5 min, 15 min, 30 min, 45 min and 60 min (runs 2 and 10–13; Table 4), with the reaction temperature being kept at 40 °C, and the Al:Ni molar ratio, at 500:1. The examination of the data revealed that the catalyst, in this case, exhibited a relatively long induction period before the active species was fully generated, with the highest activity (13.6 × 10^6^ g(PE) of (mol of Ni)^−1^ h^−1^) being seen after 15 min (run 11; Table 4). Once reached, the activity progressively declined, reaching the low point of 6.43 × 10^6^ g(PE) of (mol of Ni)^−1^ h^−1^ at the one-hour mark. Despite this drop in performance, the level remained considerable even after one hour, indicating that the active species displayed good stability and an appreciable catalytic lifetime. As seen in **Ni2**/EtAlCl_2_, the molecular weight of the polymer formed using **Ni2**/Et_2_AlCl increased with time (Appendix A) and indeed in a manner akin to that previously reported [38,40]. 

With an optimum set of conditions for **Ni2**/Et_2_AlCl now established (Al:Ni molar ratio = 500:1, run temp. = 40 °C and run time = 30 min), the three remaining nickel precatalysts (**Ni1**, **Ni3** and **Ni4**) were screened using these conditions; the full set of data is presented in Table 4 (runs 2 and 16–18; Table 4). Collectively, all precatalysts showed moderate to high activity (range: 0.04–10.7 × 10^6^ g(PE) of (mol of Ni)^−1^ h^−1^) with the relative order being **Ni2** (cyclohexyl) > **Ni1** (cyclopentyl) > **Ni4** (cyclododecyl) > **Ni3** (cyclooctyl). This trend in performance mirrors that seen when using EtAlCl_2_, with the steric properties and ring flexibility of the *ortho*-cycloalkyl group being again influential on the catalytic activity. In addition, it can be seen that the molecular weight of polyethylene produced with **Ni1**–**Ni4** varied considerably, with values ranging from 3.69 to 8.85 × 10^5^ g mol^−1^, with cyclohexyl-containing **Ni2** once more generating the highest value (Appendix A). As with EtAlCl_2_ as activator, this class of catalyst again demonstrated good control, as evidenced by the narrow dispersity of the polymer (*M_w_*/*M_n_* range: 2.1–2.5).

#### 2.2.4. Ethylene Polymerization Achieved with Activation of **Ni1**–**Ni4** with MAO

By employing an approach similar to that described for **Ni2**/EtAlCl_2_ and **Ni2**/Et_2_AlCl, **Ni2**/MAO was initially optimized to establish an effective set of reaction parameters that could be applied to **Ni1**, **Ni2** and **Ni3**; the full set of results is shown in Table 5.

In terms of temperature response (runs 1–5; Table 5), a peak in the activity of **Ni2**/MAO of 13.2 × 10^6^ g(PE) of (mol of Ni)^−1^ h^−1^ was seen at 40 °C, which then steadily decreased as the temperature was further raised. Notably, this optimum run temperature was also seen with **Ni2**/Et_2_AlCl (*cf.* 30 °C with **Ni2**/EtAlCl_2_), but with **Ni2**/MAO displaying a significantly higher level of performance. Nonetheless, **Ni2**/MAO still maintained high activity at 60 °C (7.82 × 10^6^ g(PE) of (mol of Ni)^−1^ h^−1^) and 70 °C (6.54 × 10^6^ g(PE) of (mol of Ni)^−1^ h^−1^). With regard to the molecular weight of the polymer, this was found to decline with an increase in temperature (*M_w_*: from 10.1 × 10^5^ g mol^−1^ at 30 °C to 5.75 × 10^5^ g mol^−1^ at 70 °C), while the dispersity (*M_w_*/*M_n_* range: 1.9–2.5) remained narrow; the corresponding GPC traces are shown in Appendix A. As noted in the investigations with EtAlCl_2_ and Et_2_AlCl, the rate of chain transfer increased as the temperature increased, leading to the partial deactivation of the active species. 

With the run temperature being maintained at 40 °C, the effect of the Al:Ni molar ratio on **Ni2**/MAO was investigated by increasing it from 1500:1 to 2500:1 (runs 2 and 6–9; Table 5). The scrutiny of the data revealed peak performance to have occurred at 2000:1 (13.2 × 10^6^ g(PE) of (mol of Ni)^−1^ h^−1^), above which the level of activity gently decreased to 12.0 × 10^6^ g(PE) of (mol of Ni)^−1^ h^−1^ at 2500:1. On the other hand, the activity seen at 1500:1 was notably lower (9.50 × 10^6^ g(PE) of (mol of Ni)^−1^ h^−1^), highlighting the critical number equivalents of MAO required to fully activate **Ni2**; similar observations have been noted elsewhere [20,34]. With respect to the molecular weight of polyethylene, the variation in the Al:Ni molar ratio did not have a clear impact, with values ranging from 9.33 × 10^5^ g mol^−1^ to 10.9 × 10^5^ g mol^−1^ (Appendix A); once again, the dispersity was generally narrow (*M_w_*/*M_n_* range: 2.0–2.4). 

Run time effects were also investigated using **Ni2**/MAO with separate polymerization reactions conducted over 5 min, 15 min, 30 min, 45 min and 60 min, with the run temperature and Al:Ni molar ratio being kept at 40 °C and 2000:1, respectively (runs 2 and 10–13; Table 5). As noted in **Ni2**/EtAlCl_2_, maximum polymerization activity was observed after 5 min (13.8 × 10^6^ g(PE) of (mol of Ni)^−1^ h^−1^), followed by a gradual decrease, reaching a minimum value of 10.7 × 10^6^ g(PE) of (mol of Ni)^−1^ h^−1^ after 60 min. Clearly, the active species formed quickly following the addition of MAO, which is dissimilar to the 15 min induction period observed with Et_2_AlCl. Furthermore, the relatively modest loss in activity (~22%) over time indicates that the active catalyst displayed a substantial lifetime but slightly less than the ~15% loss seen using **Ni2**/EtAlCl_2_. With respect to the molecular weight of the polymer, it increased from 7.24 × 10^5^ g mol^−1^ after 5 min to 9.68 × 10^5^ g mol^−1^ at 30 min; then, between 45 and 60 min, an unexpected reduction in weight was seen (from 8.69 × 10^5^ g mol^−1^ to 7.78 × 10^5^ g mol^−1^). The corresponding GPC traces are shown in Appendix A. The origin of this variation is uncertain but may have plausibly been due to the regeneration of active species having occurred during polymerization. Nonetheless, the dispersity of the polymer remained narrow across the different run times (*M_w_*/*M_n_* range: 1.9–2.0). 

With the optimal conditions in place for **Ni2**/MAO (Al:Ni molar ratio = 2000:1, run temp. = 40 °C and run time = 30 min), **Ni1**, **Ni3** and **Ni4** were then evaluated similarly. All the complexes showed moderate to high activity (range: 0.52–13.2 × 10^6^ g(PE) of (mol of Ni)^−1^ h^−1^), with the relative order being **Ni2** (cyclohexyl) > **Ni1** (cyclopentyl) > **Ni4** (cyclododecyl) > **Ni3** (cyclooctyl). As before, the ring size and flexibility of the cycloalkyl group proved crucial, with peak performance being again seen in the six-membered ring. As noted using the other activators, the molecular weight of polyethylenes produced with **Ni1**–**Ni4** varied considerably, with values ranging from 8.22 × 10^5^ g mol^−1^ to 15.0 × 10^5^ g mol^−1^ (Appendix A). However, in this case, the larger-ring precatalysts, **Ni3** (cyclooctyl) and **Ni4** (cyclododecyl), formed the highest-molecular-weight polymers. Nevertheless, all nickel catalysts produced polymers with narrow dispersity (*M_w_*/*M_n_* range: 2.0–2.1). 

As is evident from the three studies conducted at P_C2H4_ = 10 atm using EtAlCl_2,_ Et_2_AlCl or MAO as activator, the most productive catalyst was *ortho*-cyclohexyl-containing **Ni2** in combination with MAO (run 2; Table 5). To explore how ethylene pressure affected this performance, runs were additionally performed at 5 atm and 1 atm. At 5 atm ethylene, the activity of **Ni2**/MAO decreased dramatically from 13.2 × 10^6^ g(PE) of (mol of Ni)^−1^ h^−1^ to 3.12 × 10^6^ g(PE) of (mol of Ni)^−1^ h^−1^ (run 14; Table 5), while the molecular weight of the polymer dropped from 9.68 × 10^5^ g mol^−1^ to 7.87 × 10^5^ g mol^−1^. Further reducing the pressure to 1 atm saw the activity decrease to 0.62 × 10^6^ g (PE) of (mol of Ni)^−1^ h^−1^ (run 15; Table 5), and the molecular weight of the polymer, to 3.53 × 10^5^ g mol^−1^. Likewise, comparable pressure effects were noted with EtAlCl_2_ (see runs 2, 15 and 16; Table 3) and Et_2_AlCl (see runs 2, 14 and 15; Table 4). Evidently, a critical value of ethylene pressure is also required to achieve good performance with this class of catalysts. 

To further explore the influence of the *ortho*-cycloalkyl group, we show together activity and molecular weight data for **Ni2** alongside those on four structurally related nickel(II) bromide precatalysts (**B1**, **B4**, **B5** and **C**; see [Fig molecules-28-04852-ch001]) in Figure 3 [22,23,29]. The examination of the bar chart revealed that the catalytic activity values of **B1**, **B4**, **B5** and **C** were significantly lower than that observed with **Ni2**. It would seem likely that the reduced steric properties of **B1**, **B4** and **B5**, as a result of their particular *ortho*-substituents, were a contributing factor. On the other hand, the more excessive hindrance provided by the 2,6-dibenzhydryl substitution pattern in **C** also had a detrimental effect on activity. By contrast, this enhanced steric protection present in **C** had the effect of generating the highest-molecular-weight polyethylene, as these groups most effectively blocked chain transfer. Similarly for **Ni2**, the combination of benzhydryl and cyclohexyl as the *ortho*-substituents also served to effectively protect the metal center, leading to polyethylene of molecular weight that exceeds that produced using **B1**, **B4** and **B5**.

### 2.3. Properties of Polyethylene

According to the melting point (*T_m_*) data presented in Table 3, Table 4 and Table 5, polyethylenes exhibited values that fell, in most cases, below 100 °C ((*T_m_* range: 62.8–103.4 °C (EtAlCl_2_); 51.9–87.8 °C (Et_2_AlCl); 49.4–91.1 °C (MAO)). Such temperatures are typical of branched polyethylene, and the variations influenced by their specific branching composition and polymer molecular weight. In order to gain more detailed information on polyethylene branching, we firstly explored the influence of reaction temperature and activator on the polymers generated using **Ni2**. Secondly, the mechanical properties of selected polymer samples were investigated.

#### 2.3.1. Analysis of Polyethylene Branching with ^13^C NMR Spectroscopy

With the aim to gain some information on branching density and composition, the ^13^C NMR spectra of three representative samples, PE30_E|**Ni2**_ (run 2; Table 3), PE40_D|**Ni2**_ (run 2; Table 4) and PE40_M|**Ni2**_ (run 2; Table 5), were recorded. To allow suitable solubility, these polymer samples were dissolved in C_6_D_4_Cl_2_ at high temperature, and their spectra were recorded at 110 °C. On the basis of the chemical shift, the signal characteristics of the specific carbon environment within the branched polymer chain can be assigned and semi-quantified using methods documented in the literature [41,42,43,44,45]. The main results of this branching analysis are presented in Table 6, while the corresponding spectra are presented in Figure 4, Figure 5 and Figure 6. 

Regarding the ^13^C NMR spectrum of PE30_E|**Ni2**_, the analysis revealed 73 branches per 1000 carbons, which comprised methyl (81.8%), ethyl (2.13%), propyl (4.79%), butyl (4.03%), pentyl (2.11%) and longer branched chains (5.17%) (Figure 4). Conversely, the spectrum of PE40_D|**Ni2**_ showed relatively high branching density, containing 104 branches per 1000 carbons, including methyl (81.1%), ethyl (1.18%), propyl (3.93%), butyl (3.54%), pentyl (2.97%) and longer branches (7.23%) (Figure 5). By comparing these two polymer samples, it is evident that the content of the longer chain branches increased from 5.17% to 7.23%. On the other hand, for PE40_M|**Ni2**_, the branching density of 99 branches per 1000 carbons was determined based on methyl (82.9%), ethyl (2.53%), propyl (6.01%), butyl (3.31%) and longer branched chains (5.29%) (Figure 6). 

The above results clearly show that all three polyethylene samples contained a significant number of branches, which can be expected based on the chain-walking mechanism [46,47,48,49]. Moreover, both PE40_D|**Ni2**_ and PE40_M|**Ni2**_ showed higher branching density than PE30_E|**Ni2**_, but with a propensity for forming short-chain branches (>81%), which is corroborated by the relatively low *T_m_* values. These methyl branches can be formed with a single-chain-walking process comprising *β*-H elimination, 2,1-insertion and ethylene coordination/insertion [31,34,50,51]. While the variation in aluminum activator may be a contributing factor in this, differences in polymerization run temperatures are likely responsible. Indeed, similar findings have been previously noted [50,51], whereby higher reaction temperatures increase the degree of branching of polyethylene.

#### 2.3.2. Mechanical Performance of Branched Polyethylenes

To study the mechanical properties of the branched polyethylene produced with **Ni2** at distinct polymerization run temperatures of 30 °C and 60 °C, six polyethylene samples (PE30_E|**Ni2**_, PE30_D|**Ni2**_, PE30_M|**Ni2**_, PE60_E|**Ni2**_, PE60_D|**Ni2**_ and PE60_M|**Ni2**_) prepared using EtAlCl_2_, Et_2_AlCl and MAO were selected, and their properties were examined.

In the first instance, tensile stress–strain data were measured for all six samples (Table 7). In order to maintain consistency among the tests, each sample was divided into five parts to allow us to perform multiple tests; the stress–strain curves are shown in Figure 7, Figure 8 and Figure 9. For the two samples prepared using EtAlCl_2_ as activator, the stress–strain curves are presented in Figure 7. PE30_E|**Ni2**_ was observed to exhibit higher ultimate tensile stress (6.98 MPa) than PE60_E|**Ni2**_ (4.29 MPa), with correspondingly higher crystallinity (*X*_c_ = 16.2% vs. 4.3%). By contrast, PE30_E|**Ni2**_ exhibited lower strain at break (*ε_b_* = 353%) than PE60_E|**Ni2**_ (*ε_b_* = 462%). Similar results were observed for the two samples prepared using **Ni2**/Et_2_AlCl (Figure 8), that is PE60_D|**Ni2**_ showed lower tensile stress (8.59 MPa) and crystallinity (*X*_c_ = 5.5%), as well as higher strain-at-break (*ε_b_* = 861%), than PE30_D|**Ni2**_. This was also the case for the samples prepared using **Ni2**/MAO (Figure 8), among which PE30_M|**Ni2**_ exhibited a higher tensile stress (15.91 MPa) and crystallinity (22.3%) and lower strain at break (713%) than PE60_M|**Ni2**_. 

Overall, these results indicate that crystallinity influenced the tensile properties of these polyethylenes [53], with high-crystallinity polyethylene displaying high ultimate tensile strength but low strain at break. In addition, when comparing the data among samples, the difference in the type of activator also influenced the mechanical properties. For example, the ultimate tensile stress of PE30_E|**Ni2**_ was 6.98 MPa, which compares to 15.91 MPa for PE30_M|**Ni2**_, while the strain at break value of 353% compared to 713%, respectively. Notably, when comparing these test data with those of previously reported polyethylene produced under similar conditions (i.e., Et_2_AlCl or MAO as activator, P_C2H4_ = 10 atm and run temp. = 60 °C), PE60_E|**Ni2**_ and PE60_M|**Ni2**_ exhibited higher values of tensile stress and strain at break [37,38,54].

Secondly, to evaluate the capacity for elastic recovery of these polyethylenes, the six samples were tested for hysteresis at 30 °C using dynamic mechanical analysis; the stress–strain response curves of each sample are shown in Figure 10, Figure 11 and Figure 12. The strain recovery value (SR) was calculated with the standard equation SR = 100 (*ε_a_* − *ε_r_*)/*ε_a_* (where *ε_a_* is the applied strain and *ε_r_* is the cyclic strain at zero load after 10 cycles). Each sample was tested over 10 cycles, and elastic recovery was set at approximately 80% of the maximum tensile strength. The inspection of Figure 10, Figure 11 and Figure 12 reveals that all samples displayed a constant level of recovery after the first cycle. By comparing the data of the samples produced at different run temperatures using the same activator, it is evident that the materials produced at a higher run temperature, 60 °C, had a higher elastic recovery rate (61.6% for PE60_E|**Ni2**_; 60.5% for PE60_D|**Ni2**_; 71.2% for PE60_M|**Ni2**_) than those produced at the lower run temperature of 30 °C (47.4% for PE30_E|**Ni2**_; 48.9% for PE30_D|**Ni2**_; 47.9% for PE30_M|**Ni2**_). In effect, these data indicate that apart from crystallinity, the operating temperature of the polymerization and in turn the polymer molecular weight had a significant effect on the elastic properties of polyethylene. As a further notable point, the elastic recovery properties of these polymer samples were higher than those reported in previous studies of nickel catalysts [38,55,56].

## 3. Materials and Methods

### 3.1. General Considerations

All reactions involving air- and moisture-sensitive compounds were conducted in a strict water- and oxygen-free environment (nitrogen purity > 99.9%). Under a nitrogen atmosphere (>99.9%), toluene was firstly dried over sodium/benzophenone ketyl and then distilled immediately prior to use. Methylaluminoxane (MAO; 1.45 M in toluene) and modified methylaluminoxane (MMAO; 2.50 M in heptane) were purchased from Anhui Botai Electronic Materials Co., Ltd (Anhui, China). Ethylaluminum dichloride (EtAlCl_2_), ethylaluminum sesquichloride (EASC) and diethylaluminum chloride (Et_2_AlCl) were bought from Lianli Chemical (Beijing, China), while high-purity ethylene was acquired from Beijing Yanshan Petrochemical Co (Beijing, China). Other reagents and solvents were purchased from Acros, Aldrich (Beijing, China) or other local suppliers and were used upon receipt. The FT-IR spectrum data of the complexes were recorded at room temperature with a Nicolet 6700 FT-IR spectrometer (Thermo fisher, Shanghai, China) in the region 4000–400 cm^−1^, and the relative intensity of the IR bands was described as w (weak), m (medium) or s (strong). The elemental analysis was performed with a FlashEA 1112 microanalyzer (Thermo Flash Smart, Beijing, China). The melting temperatures of the polyethylene samples were measured under nitrogen during the fourth scan with a PerkinElmer TA-Q2000 (Beijing, China) differential scanning calorimeter. The molecular weight (*M*_w_) and dispersity (*M*_w_/*M*_n_) of the polymers were determined using an Agilent PL-GPC 220 instrument (Beijing, China) by employing 1,2,4-trichlorobenzene as the eluting solvent at an operating temperature of 160 °C. A Bruker AVANCE III 500 MHz instrument (Beijing, China) at 110 °C was used to record the ^13^C NMR spectra of the polyethylene samples. Sample preparation consisted in dissolving 60–80 mg of polyethylene samples in 1,2-dichlorobenzene-*d*_4_ (2 mL), using TMS as an internal standard. 2-Cyclopentyl-4,6-dibenzhydrylaniline, 2-cyclohexyl-4,6-dibenzhydrylaniline, 2-cyclooctyl-4,6-dibenzhydrylaniline and 2-cyclododecyl-4,6-dibenzhydrylaniline were prepared using the corresponding procedure detailed in the literature [57,58,59].

### 3.2. Synthesis of [ArN=C(Me)-C(Me)=NAr]NiBr_2_ (**Ni1**–**Ni4**)

(a)Ar = 2-(C_5_H_9_)-4,6-(CHPh_2_)_2_C_6_H_2_ (**Ni1**). 2,3-Butanedione (0.009 g, 0.10 mmol), 2-cyclopentyl-4,6-dibenzhydrylaniline (0.198 g, 0.40 mmol) and (DME)NiBr_2_ (0.031 g, 0.10 mmol) were stirred and heated to reflux in glacial acetic acid (15 mL) for 6 h. After cooling to room temperature, excess anhydrous diethyl ether was added to induce precipitation. The precipitate was then filtered, washed with diethyl ether (4 × 20 mL) and dried under reduced pressure to give **Ni1** (0.021 g, 17%) as brown powder. FT-IR (cm^−1^): 2954(w), 2869(w), 1644(w, v_C=N_), 1578(m), 1494(m), 1448(m), 1418(w), 1340(w), 1265(w), 1215(w), 1186(w), 1163(w), 1077(w), 1031(w), 745(m), 699(s). Anal. Calcd for C_78_H_72_Br_2_N_2_Ni (1255.95): C, 74.59; H, 5.78; N, 2.23. Found: C, 74.23; H, 6.01; N, 2.04.(b)Ar = 2-(C_6_H_11_)-4,6-(CHPh_2_)_2_C_6_H_2_ (**Ni2**). Using a synthetic procedure similar to that described for **Ni1** but using 2-cyclohexyl-4,6-dibenzhydrylaniline as the arylamine, **Ni2** was isolated as brown powder (0.327 g, 64%). FT-IR (cm^−1^): 2921(w), 2850(w), 1637(w, v_C=N_), 1600(m), 1495(m), 1446(m), 1368(m), 1260(w), 1239(w), 1215(w), 1162(w), 1077(w), 1032(w), 776(w), 748(m), 699(s). Anal. Calcd for C_80_H_76_Br_2_N_2_Ni (1284.00): C, 74.83; H, 5.97; N, 2.18. Found: C, 74.67; H, 6.22; N, 1.94.(c)Ar = 2-(C_8_H_15_)-4,6-(CHPh_2_)_2_C_6_H_2_ (**Ni3**). Using a synthetic procedure similar to that described for **Ni1** but using 2-cyclooctyl-4,6-dibenzhydrylaniline as the arylamine, **Ni3** was isolated as brown powder (0.068 g, 20%). FT-IR (cm^−1^): 2919(w), 2857(w), 1644(w, v_C=N_), 1593(m), 1494(m), 1447(m), 1421(w), 1345(w), 1185(w), 1078(w), 1030(w), 914(w), 849(w), 744(m), 698(s). Anal. Calcd for C_84_H_84_Br_2_N_2_Ni (1340.11): C, 75.29; H, 6.32; N, 2.09. Found: C, 74.98; H, 6.44; N, 1.83.(d)Ar = 2-(C_12_H_23_)-4,6-(CHPh_2_)_2_C_6_H_2_ (**Ni4**). Using a similar synthetic procedure to that described for **Ni1** but using 2-cyclododecyl-4,6-dibenzhydrylaniline as the arylamine, **Ni4** was isolated as brown powder (0.072 g, 25%). FT-IR (cm^−1^): 2920(w), 2858(w), 1639(w, v_C=N_), 1594(m), 1494(m), 1445(m), 1373(w), 1258(w), 1212(w), 1161(w), 1075(w), 1031(w), 914(w), 847(w), 743(m), 697(s). Anal. Calcd for C_92_H_100_Br_2_N_2_Ni (1452.33): C, 76.09; H, 6.94; N, 1.93. Found: C, 75.83; H, 7.03; N, 1.77.

### 3.3. X-ray Diffraction Studies

Single crystals of **Ni2** and **Ni4** were grown by diffusing diethyl ether into a dichloromethane solution of each complex. Selected crystals of each were mounted on an XtaLAB SynergyR single-crystal diffractometer. Cell parameters were obtained by performing the global refinement of the positions of all collected reflections with monochromatic Cu-Kα radiation (λ = 1.54184 Å) at a temperature of 170.01(10) K. The intensity was corrected for Lorentz and polarization effects, as well as empirical absorption. The structure was determined using the SHELXT (Sheldrick, 2015) [60] direct method (and Patterson’s method) and refined with full matrix least squares for F^2^ based on SHELXL (Sheldrick, 2015) [61]. All hydrogens were placed in the calculated positions, and all non-hydrogen atoms were refined with anisotropic displacement parameters. The Squeeze option in the crystallographic program PLATON was applied to remove disordered solvents from the structure, and possible twin structures were searched with TwinRotMat [62,63]. The graphical representation of the molecular structure of both complexes was generated with the ORTEP application [64]. X-ray structure determination and detailed data are provided in Appendix A.

### 3.4. Polymerization Experiments

Ethylene polymerization runs were carried out in a stainless-steel, 250 mL capacity autoclave equipped with a mechanical stirrer and a temperature controller. The autoclave was evacuated and backfilled with nitrogen three times and then with ethylene once. Freshly distilled toluene (25 mL) was injected into the autoclave under an ethylene atmosphere. After the temperature had stabilized, a solution of the precatalyst (1 μmol) in toluene (50 mL) was added, followed by a pre-determined amount of activator and finally some more toluene (25 mL). At the designated ethylene pressure, the reaction mixture was stirred (400 rpm/min) for the selected time. Once the run was completed, the reactor was allowed to cool to room temperature. The pressure within the vessel was slowly released, and the contents were acidified with ethanol to quench the polymerization. After a period of stirring, the polymer was collected using filtration and washed with ethanol. The product was dried under reduced pressure at 60 °C and then weighed.

## 4. Conclusions

In summary, four α-diimine–nickel(II) complexes bearing four distinct *ortho*-cycloalkyl substituents, namely, cyclopentyl (**Ni1**), cyclohexyl (**Ni2**), cyclooctyl (**Ni3**) and cyclododecyl (**Ni4**), were successfully prepared using a simple one-pot synthetic route. All the complexes were well characterized, and in the cases of **Ni2** and **Ni4**, this was performed additionally using single-crystal X-ray diffraction. When activated with either EtAlCl_2_, Et_2_AlCl or MAO, **Ni1**–**Ni4** displayed moderate to very high activity as catalysts for the polymerization of ethylene, forming polyethylene with high molecular weight, high branching density and a distinct preference for short-chain methyl branches (>80%). In particular, cyclohexyl-containing **Ni2**, following activation with MAO, was the most productive of this work, exhibiting a peak activity of 13.2 × 10^6^ g(PE) of (mol of Ni)^−1^ h^−1^ at 40 °C. Furthermore, this MAO-activated nickel catalyst formed the highest-molecular-weight polymer (9.68 × 10^5^ g mol^−1^) when compared with those obtained using **Ni2**/EtAlCl_2_ (7.51 × 10^5^ g mol^−1^) and **Ni2**/Et_2_AlCl (8.85 × 10^5^ g mol^−1^) at the same run temperature. Overall, the ring size of the *ortho*-cycloalkyl group displayed a distinct effect on catalyst activity, with **Ni2** (cyclohexyl) > **Ni1** (cyclopentyl) > **Ni4** (cyclododecyl) > **Ni3** (cyclooctyl). By contrast, its correlation with polymer molecular weight was less clear and dependent on the aluminum activator employed. Nonetheless, all catalysts showed good control, with narrow dispersity being a feature of all polymeric materials produced. In terms of the mechanical properties of these macromolecules, it was evident that the degree of crystallinity (*X*_c_) and molecular weight (*M*_w_, linked to run temperature) were influential factors in tensile strength, strain at break and elastic recovery. Further studies to explore the applications of these thermoplastic elastomers (TPEs) are ongoing.

## Data Availability

Not applicable.

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
