# Peer review of "N,N-Bis(2,4-Dibenzhydryl-6-cycloalkylphenyl)butane-2,3-diimine–Nickel Complexes as Tunable and Effective Catalysts for High-Molecular-Weight PE Elastomersâ€"

_molecules, 2023, doi:10.3390/molecules28124852_

Round 1
Reviewer 1 Report
Authors - Shu Jiang, Yuting Zheng, Irina V. Oleynik, Zhixin Yu, Gregory A. Solan, Ivan I. Oleynik, Ming Liu, Yanping Ma, Tongling Liang and Wen-Hua Sun
Manuscript « N,N-bis(2,4-Dibenzhydryl-6-cycloalkylphenyl)butane-2,3-diimine-nickel complexes as tunable and effective catalysts for high molecular weight PE elastomers»
I would like to note that the work was done at a high professional level. The conditions of catalytic polymerization of ethylene have been thoroughly studied and described in detail, which will obviously allow scaling the results obtained. The results of the research are beyond doubt. The work undoubtedly deserves a quick publication.
But at the same time there were questions about the nickel complex.
1. The resulting nickel complexes were the only individual products in the reaction? The authors do not provide information about how they cleaned the complexes, from which solvents they crystallized. Since there is an X-ray of single crystals, I would like to understand in which solvent they grow. Also, the IR data is not enough to fully characterize the complexes. There are many examples where such complexes are characterized by nuclear magnetic resonance and mass spectrometry (ESI). Or the catalysts were not obtained for the first time, then it is required to give references?
2. There was also a question about loading the catalyst by weight into the reactor. There was also a question about loading the catalyst by weight into the reactor. How many grams of catalyst and activator were taken, if their ratio is 1:500?
Author Response
The response to the reviewer's comments has been uploaded.

Author Response

(The authors gave the same response as above.)
